# Impact of Ce/Zr Ratio in the Nanostructured Ceria and Zirconia Composites on the Selective CO_2_ Adsorption

**DOI:** 10.3390/nano13172428

**Published:** 2023-08-26

**Authors:** Gloria Issa, Martin Kormunda, Oyundari Tumurbaatar, Ágnes Szegedi, Daniela Kovacheva, Daniela Karashanova, Margarita Popova

**Affiliations:** 1Institute of Organic Chemistry with Centre of Phytochemistry, Bulgarian Academy of Sciences, 1113 Sofia, Bulgaria; gloria.issa@orgchm.bas.bg (G.I.); oyundari.tumurbaatar@orgchm.bas.bg (O.T.); 2Faculty of Science, University of Jan Evangelista Purkyně, Pasteurova 3632/15, 400 96 Ústí nad Labem, Czech Republic; martin.kormunda@ujep.cz; 3Research Centre for Natural Sciences, Institute of Materials and Environmental Chemistry, Magyar Tudosok krt. 2, 1117 Budapest, Hungary; szegedi.agnes@ttk.hu; 4Institute of General and Inorganic Chemistry, Bulgarian Academy of Sciences, 1113 Sofia, Bulgaria; didka@svr.igic.bas.bg; 5Institute of Optical Materials and Technologies, Bulgarian Academy of Sciences, 1113 Sofia, Bulgaria; dkarashanova@yahoo.com

**Keywords:** CO_2_ capture, CeO_2_, ZrO_2_, Ce-Zr composite nanoparticles

## Abstract

High surface-area, mesoporous CeO_2_, ZrO_2,_ and Ce-Zr composite nanoparticles were developed using the hydrothermal template-assisted synthesis method. Samples were characterized using XRD, N_2_ physisorption, TEM, XPS, and FT-IR spectroscopic methods. The CO_2_ adsorption ability of the obtained materials was tested under dynamic and equilibrium conditions. A high CO_2_ adsorption capacity in CO_2_/N_2_ flow or CO_2_/N_2_/H_2_O was determined for all studied adsorbents depending on their composition flow. A higher CO_2_ adsorption was registered for Ce-Zr composite nanomaterials due to the presence of strong O^2−^ base sites and enriched surface oxygen species. The role of the Ce/Zr ratio is the process of the formation of highly active and selective adsorption sites is discussed. The calculated heat of adsorption revealed the processes of chemisorption and physisorption. Experimental data could be appropriately described by the Yoon–Nelson kinetic model. The composites reused in five adsorption/desorption cycles showed a high stability with a slight decrease in CO_2_ adsorption capacities in dry flow and in the presence of water vapor.

## 1. Introduction

Today’s rapid economic development contributes to a significant increase in the use of energy obtained from conventional fossil fuels (coal, oil, and natural gas) [1,2]. The massive use of fossil fuels leads to adverse effects on the environment, namely global warming, and critical climate change on earth. Global warming is the result of the increasing concentration of greenhouse gases and in particular, carbon dioxide (CO_2_), the main anthropogenic greenhouse gas [3]. The significant increase in CO_2_ concentration in the atmosphere from 340 ppm in 1980 to 408 ppm in 2019 leads to a negative impact on the environment [4]. Nowadays, the high concentration of carbon dioxide in the atmosphere can be reduced by increasing the share of renewable carbon sources or reducing CO_2_ emissions with anthropogenic origin [1]. For the elimination of greenhouse gases, much attention has been focused on improving the activity of absorbents for the capture of CO_2_. Various CO_2_ capture technologies are available at present including adsorption, absorption (chemical and physical absorptions), and different membrane technologies [5,6,7,8,9,10]. The development of efficient materials for these processes is of great importance. Therefore, the CO_2_ adsorption efficiency can be improved by selecting an appropriate material as an adsorbent. At present, a lot of porous material hosts, such as activated carbon, zeolites, organic polymers, mesoporous silicas, metal-oxide molecular sieves, finely dispersed CaO and nanosized metal-oxide composites are widely investigated [11,12,13,14,15,16,17,18,19]. Among these materials, mesoporous oxide-based adsorbents have been thoroughly studied in recent years. They can be promising candidates in CO_2_ capture applications due to the possibility to control the pore structure, their good chemical/thermal stability, and good mass transfer of modifiers to the mesoporous matrix. Particular interest and efforts have been focused on the preparation of porous metal oxides with a high specific surface area consisting of crystalline nanosized particles, using various synthesis techniques [13]. The main advantage of these nano-systems over conventional bulk oxides is the large number of available and easily accessible active centers due to the nanoscale state of the material and the very well-developed outer surface.

The presence of water in the flue gas from thermal power plants is one of the major problems for selective CO_2_ separation by adsorbents [14]. A significant decrease in CO_2_ adsorption in the presence of H_2_O is observed for conventional adsorbents such as zeolite. The selectivity for CO_2_ capture can be increased by applying metal oxides as adsorbents assuming that H_2_O is physically adsorbed onto their surface and CO_2_ is chemically adsorbed [14]. Among these oxides, CeO_2_ exhibited a larger adsorption amount (around 275 mmol/L in dry conditions at 50 °C) and relatively low desorption temperature (below 190 °C) [14]. A remarkable improvement in its physicochemical properties such as density, ionic conductivity, thermal properties, and lattice parameters [15]. The cerium–zirconium oxide systems are used as oxygen storage materials and they are excellent environmental catalysts for pollutant removal, showing high activity, selectivity, and stability [20,21]. Various preparation procedures have been developed for the preparation of mixed cerium–zirconium oxide materials, such as co-precipitation, micro-emulsion methods, sol-gel, thermal method, template synthesis, etc. [22,23,24,25,26,27,28]. Hsiang et al. [22] used the co-precipitation method and found that Ce_0.6_Zr_0.4_O_2_ decomposed into Ce-rich (cubic structure) and Zr-rich (tetragonal structure) phases from a single cubic phase after the calcination process at high temperature. Rumruangwong et al. [24] synthesized ceria–zirconia mixed oxide by sol-gel technique and established that the surface area of the Ce_x_Zr_1−x_O_2_ powders was improved by increasing ceria content, and their thermal stability was increased by the incorporation of ZrO_2_. Over the past several years, different surfactants (CTAB, Pluronic P123, and F127) have been applied as templates to synthesize mesoporous nanosized ceria–zirconia materials [27,29]. The removal of surfactants after calcination at high temperatures gave rise to the formation of fluorite-structured Ce_x_Zr_1−x_O_2_ materials with well-developed mesoporous structures and a high specific surface area. It has been established that the Ce–Zr mixed oxides are more active catalysts than pure CeO_2_, due to the partial substitution of Ce^4+^ (ionic radii = 0.97 Å) with Zr^4+^ (ionic radii = 0.84 Å), which leads to deformation of the lattice, improving its oxygen storage capacity, the redox properties, and enhancing the catalytic performance and thermal stability [30,31,32,33,34,35,36]. Suguira Masahiro et al. [37] demonstrated that the incorporation of the significantly smaller Zr^4+^ ion into the cerium-oxide lattice leads to the formation of oxygen vacancies associated with structural relaxation, due to the reduction of Ce^4+^ to the larger Ce^3+^ ions. The structural, textural, and reduction properties of Zr-based catalysts can be ascribed to the higher number of defects, formed upon the addition of Zr^4+^ ions into the ceria lattice, and the subsequent increase in oxygen mobility [38]. The authors [38] established that the surface oxygen sites adjacent to the Ce(III) favor CO_2_ adsorption compared to those adjacent to Ce(IV), Zr, or surface hydroxyl sites. However, CO_2_ adsorption capacity and CO_2_ desorption trends in the presence of H_2_O for metal-oxide adsorbents remained unclear despite its importance, considering that flue gas contains larger amount of H_2_O.

In the present study, we focused our attention on the preparation of Ce–Zr composite materials in a wide range of compositions using template-assisted hydrothermal synthesis. The pure CeO_2_, pure ZrO_2,_ and Ce–Zr composites were studied in CO_2_ adsorption experiments. The comparative study of samples with different Ce/Zr ratios and a variation of preparation procedures was used for understanding the impact of structural, morphological, textural, and surface features of these materials on their CO_2_ adsorption capacity, which is important for the control and optimization of the adsorbents’ properties.

## 2. Experimental Section

### 2.1. Materials

Cerium(III) chloride heptahydrate (99%) (Alfa Aesar, Kendal, Germany), zirconium chloride anhydrous (98%) (Sigma-Aldrich, Saint Louis, MO, USA), and hexadecyltrimethylammonium bromide (CTAB) (≥98%) (Sigma-Aldrich Chemie, Schnelldorf, Germany) chemicals were used without further purification.

### 2.2. Preparation of ZrO_2_, CeO_2_ and Ce_x_Zr_y_ Adsorbents

Ce_x_Zr_y_ materials were synthesized via hydrothermal method at Ce:Zr molar feed ratios of 1:1 (Ce_0.5_Zr_0.5_), 1:2 (Ce_0.33_Zr_0.67_), and 2:1 (Ce_0.67_Zr_0.33_). Endmembers of the series, i.e., CeO_2_ and ZrO_2_, were prepared as well. 

#### 2.2.1. Synthesis of CeO_2_ and ZrO_2_ Adsorbents

The synthesis of CeO_2_ and ZrO_2_ was performed using the following procedure; 5.41 g cerium(III) chloride heptahydrate (CeCl_3_∙7H_2_O) or 3.75 g zirconium chloride (ZrCl_4_), respectively, were completely dissolved in 50 mL distilled water at 50 °C and stirred for 45 min. The solution was mixed with a solution containing 10 mL 25 wt.% ammonia (NH_3_) in 10 mL H_2_O. The homogeneous solution was transferred to a Teflon vessel jacketed in a stainless-steel autoclave and heated in an oven at 100 °C for 24 h under static conditions. The white precipitate product was filtered, washed with distilled water, and dried at 40 °C overnight. 

#### 2.2.2. Synthesis of Ce_0.5_Zr_0.5_, Ce_0.67_Zr_0.33_, and Ce_0.33_Zr_0.67_ Adsorbents

The synthesis of (1) Ce_0.5_Zr_0.5_, (2) Ce_0.67_Zr_0.33,_ and (3) Ce_0.33_Zr_0.67_ was performed using the following procedure. A solution of (1) 1.87 g (2) 1.25 g (3) 2.5 zirconium chloride (ZrCl_4_) in 50 mL distilled water and (1) 2.71 g, (2) 3.61 g, and (3) 1.81 g cerium(III) chloride heptahydrate (CeCl_3_∙7H_2_O) in 50 mL water were added at room temperature to a solution of 6 g CTAB in 100 mL water and stirred for 45 min. The above solution was mixed with a solution containing 10 mL ammonia 25 wt.% (NH_3_) in 10 mL water. The homogeneous solution was transferred to a Teflon vessel jacketed in a stainless-steel autoclave. Subsequently, the temperature was increased to 100 °C and aged for 24 h under static conditions. The white precipitate product was filtered, washed with distilled water, and dried at 40 °C overnight. The template was removed by calcination at 300 °C for 6 h. An adsorbent with Ce_0.5_Zr_0.5_ composition was prepared with the same procedure, but the template was removed by extraction method. The filtered synthesis product was extracted with ethanol at 80 °C for 5 h and dried under vacuum at 40 °C overnight. The latter one is denoted as ext.Ce_0.5_Zr_0.5._

### 2.3. Characterization 

Powder X-ray diffraction patterns were collected on Bruker D8 Advance diffractometer equipped with Cu Kα radiation and LynxEye detector. Phases were identified by Diffrac.EVA v.4 using ICDD-PDF-2 (2021) database. For the estimation of the mean crystallite size, the broadening of the diffraction lines was analyzed by means of whole powder pattern profile fitting using the program Topas v.4.2. The corrections for the instrumental broadening were included by the fundamental approach technique implemented in the program (accounting for the real elements on the beam path).

Determination of the specific surface area and pore size distribution was performed by low-temperature nitrogen adsorption. The adsorption and desorption isotherms of nitrogen at −196 °C were determined in the pressure range of p/p_0_ = 0.001–1, using an advanced micropore size and chemisorption analyzer “AUTOSORB iQ-MP/AG” (Anton Paar GmbH, Graz, Austria). Before every measurement, the samples were degassed at 80 °C for 16 h. 

The morphology, the phase composition, and the microstructure of the samples were analyzed by Transmission Electron Microscopy (TEM) using High Resolution Transmission Electron Microscope HRTEM JEOL JEM 2100 (JEOL Ltd., Tokio, Japan). All measurements were held at 200 kV accelerating voltage. Match software (Version 3.13). Crystal Impact GbR, Bonn, Germany) with Crystallography Open Database (COD) was applied for the phase identification of the samples. The measurements of the nanoparticles diameters for the statistical analysis were implemented by means of Image J freeware (National Institutes of Health, Bethesda, MD, USA).

X-ray photoelectron spectroscopy (XPS) spectra have been obtained using a Phoibos 100 (SPECS) based X-ray photoelectron spectrometer operating in Fixed Analyzer Transmission (FAT) mode with 5-channel MCD-5 detector (SPECS). The spectrometer is equipped with a non-monochromatic X-ray source XR50 with double Al/Mg anode operated under 12 kV (200 W), the core-level spectra were measured using Al Kα radiation (hν = 1486.6 eV), and no flood gun was used. The samples were placed on double-sided carbon conductive adhesive tape. The spectra analyses were made in CasaXPS software with the use of the Shirley background model and built-in RSF for composition calculations. The binding energy was adjusted on a base of valence band XPS spectra.

In situ FT-IR spectra were measured by a Nicolet Compact 6700 spectrometer (Thermo Fisher Scientific, Waltham, MA, USA). Thin, self-supported wafers with 2 cm^2^ surface area were prepared and pyridine (Py, 6 mbar) was adsorbed at 100 °C for 30 min on the formerly dehydrated samples (300 °C, 1 h) in a special, in situ FT-IR cell. Subsequently, Py was desorbed by evacuation at elevating temperatures 100–300 °C for 30 min. Spectra were recorded at the IR beam temperature with 128 scans at a resolution of 2 cm^−1^. For quantitative comparison, the spectra were normalized to 5 mg/cm^2^ weight.

In situ FT-IR CO_2_ adsorption experiments were conducted in the same spectroscopic system. Mixed-oxide adsorbents were pretreated at 300 °C in high vacuum for 1 h before room temperature CO_2_ adsorption (15 mbar) for 30 min. Spectra were recorded in CO_2_ atmosphere at RT, followed by evacuation at RT, 100, 200, and 300 °C. Spectra were also measured with the same parameters as before, namely 128 scans at 2 cm^−1^ resolution and were normalized to 5 mg/cm^2^ weight.

### 2.4. Dynamic and Static CO_2_ Adsorption

CO_2_ adsorption experiments were performed in dynamic conditions in a flow-through system. The sample (0.40 g adsorbent) was dried at 150 °C for 1 h, and 3 vol. % CO_2_/N_2_ at a flow rate of 30 mL/min was applied for the adsorption experiments at 25 °C. The gas was analyzed online by using a gas chromatograph NEXIS GC-2030 ATF (Shimadzu, Japan) with a 25 m PLOT Q capillary column. CO_2_ adsorption measurements under static conditions were measured at 0 °C and 25 °C with an AUTOSORB iQ-MP-AG (Anton Paar GmbH, Graz, Austria) surface area and pore size analyzer (from Quantachrome, Anton Paar GmbH, Graz, Austria). 

## 3. Results and Discussion

### 3.1. Textural Characterization

Nitrogen physisorption isotherms of the prepared adsorbents are shown in Appendix A. Specific surface area and total pore volume, calculated from the nitrogen adsorption/desorption isotherms are presented in Table 1. All the isotherms are of type IV, ZrO_2_ and CeO_2_ with H3 type, and the mixed oxides are similarly to each other, with H2 type hysteresis loops. ZrO_2_ showed the highest specific surface area and pore volume, and on the contrary, CeO_2_ has the lowest one. The Ce–Zr composite adsorbents showed transition values proportional to their zirconia content. The use of the extraction method for template removal (Table 1) resulted in a higher surface area than its counterpart obtained by calcination. The observed differences in the shape of the hysteresis loop indicate the presence of “slit-like” or wedge-shaped pores between flaky particles of pure ZrO_2_ and CeO_2_ oxides, and “ink-bottle-like” pores for all mixed-oxide materials.

XRD patterns of samples are presented in Figure 1. 

The phase composition, unit cell parameters, and average crystallite size are shown in Table 2. The XRD pattern of ZrO_2_ represents a broad peak typical of amorphous materials. The CeO_2_ adsorbent exhibits a face-centered, cubic, fluorite-type structure with crystallite size of 16 nm. In the case of the mixed metal-oxide samples, the CeO_2_ reflections are broader, lower in intensity, and slightly shifted to higher Bragg angles as well. The broadening of the reflections means a decrease in the crystallite size of the fluorite-like phase, whereas the intensity decrease together with the shifting of position could be assigned to the incorporation of Zr in the ceria lattice. The differences observed in the lattice parameters with the change of the Ce/Zr ratio suggest an alteration in the degree of incorporation of Zr into the fluorite lattice. Considering the smaller ionic radius of Zr^4+^ (0.084 nm) compared to Ce^4+^ (0.097 nm), an incorporation of the Zr into ceria lattice can be concluded [30,39,40]. A significant shift of the lattice parameter is observed for the Ce_0.33_Zr_0.67_ nanomaterial most probably due to the strongest interaction between CeO_2_ and ZrO_2_.

### 3.2. TEM Analysis

In Figure 2, TEM micrographs at low (40,000×) and high (600,000×) magnifications, histograms of the particles size distribution, and Selected Area Electron Diffraction (SAED) patterns for the composite samples are presented. 

The statistical analysis of TEM images revealed a formation of relatively uniform nanoparticles with mean sizes between 3 and 5 nm and narrow size distributions for all three materials. The calculated mean particle diameters d_mean_ for the tree samples studied with the corresponding standard deviations (SD) are as follows: d_1mean_ = 3.21 nm (SD = 0.92 nm) for Ce_0.33_Zr_0.67_, d_2mean_ = 3.53 nm (SD = 0.95 nm) for Ce_0.5_Zr_0.5_, and d_3mean_ = 4.91 nm (SD = 2.05 nm) for Ce_0.67_Zr_0.33_ sample. They follow the same trend of slightly increasing with a decreasing amount of zirconia, similarly to the average crystallite diameters obtained from X-ray diffraction. The SAED patterns and HRTEM indexing and phase composition analysis clearly proved the interaction of CeO_2_ and ZrO_2_ by the established formation of a solid solution phase (Ce,Zr)O_2_ in the mixed-oxide samples. The phase of monometallic oxide ZrO_2_ was also detected in all samples (CeO_2_ cubic, a = 5.40730 Å, COD Entry #96-156-2990; ZrO_2_ tetragonal, a = 3.61200 Å c = 5.21200 Å, COD Entry #96-152-6428).

### 3.3. X-ray Photoelectron Spectroscopy (XPS)

The XPS spectra (Figure 3) of the CeO_2_ sample shows multiple peaks at 882.5 and 900.9 eV, due to photoemission from Ce 3d3/2 and Ce3d5/2 levels for Ce^4+^ ions, respectively and the peaks at 888.7, 898.3, 907.2, and 916.7 eV are its Ce 3d Ce^4+^ satellites [41,42,43,44]. 

Ce^3+^ oxides have Ce 3d_3/2_ and Ce 3d_5/2_ spectra consisting of another two multiplets. The peaks with the highest binding energies are localized at 903.3 eV and 884.8 eV, and the energy states with low binding energy are at 899.4 eV and 880.9 eV, respectively [41]. The observed doublets in Ce 3d spectra are typical of Ce^4+^ ions and Ce^3+^ ions. The observed slight decrease in the atomic concentration of the Ce 3d component for the bi-component samples compared to pure CeO_2_ may be due to the formation of Zr–O–Ce bonds [45]. XPS spectra of the mixed-oxide samples [46,47] showed that the O 1s peak was decomposed into two sets of components.

The peaks at about 529.5 and 529.7 eV correspond to oxygen from the oxide lattice in Ce^4+^ and Zr^4+^. The observed slight shift to lower binding energy is probably due to the interaction between CeO_2_ and ZrO_2_. The third component at 531.5 eV is usually associated with chemisorbed oxygen, such as hydroxyl groups, O_2_^2−^, O^−^ oxygen centres, or molecular oxygen [46] but here it is also a component associated with oxygen bonded to Ce^3+^. This effect clearly shows that the close contact between the metals in the mixed-oxide system leads to the formation of oxygen vacancies and increased oxygen mobility. In addition, promoting the oxygen mobility leads to enhancing the basicity, and enriching the surface oxygen species, which are efficient at activating CO_2_.

Hydroxyl groups associated with a binding energy of about 533 eV were also observed in an amount decreasing from 9% for the CeO_2_ sample to less than 3% for samples with higher Zr content. The BE of Zr 3d_5/2_ and Zr 3d_3/2_ is 181.9 eV and 184.2 eV, respectively [48], and the peaks are registered at the same position in the spectra of all Zr-containing samples. The surface area elemental composition is presented in Table 3. It is obvious that the Ce^3+^ concentration on the surface is higher for the Ce_0.67_Zr_0.33_ and Ce_0.33_Zr_0.67_ mixed-oxide nanomaterials compared to pure CeO_2_ (Table 3). This is an indication of a higher degree of substitution of Zr ions in the cerium-oxide lattice and enhanced oxygen mobility. It should be noted that, the small systematic shift of binding energy of O-Ce^4+^, O-Ce^3+^ and O-Zr^4+^ peaks (Table 3), which is in accordance with X-ray analysis (Table 2) showing strong interaction between cerium and zirconium oxides, partial incorporation of smaller zirconium ions into the fluorite-oxide lattice and formation of cerium ions in a low oxidation state. The Ce^3+^ surface concentration is higher for Ce–Zr composite materials as compared to pure CeO_2_ (Table 3), which indicates a high degree of Zr incorporation into ceria lattice with the formation of oxygen vacancies [49]. XPS analysis data of high ceria containing samples (CeO_2_, Ce_0.67_Zr_0.33_) show that the concentration of Ce is lower on the surface than theoretically calculated, whereas the oxygen content is higher than the theoretical content (73/71 at. %, instead of 66 at. %). It means that these samples are surface-rich with defect sites and OH groups due to the very small particle size or adsorbed O-containing species.

The amount of surface oxygen functional groups or adsorbed O-containing species in all mixed oxides increases. The absence of zirconium oxide in a lower oxidation state is indicative of segregation of the zirconia phase over the ceria particles, which is in accordance with XRD and TEM results.

### 3.4. In Situ FT-IR Spectroscopy of Adsorbed Pyridine and CO_2_

The surface species of oxides can also be described as acid-base pairs. Oxygen atoms of the structure serve as basic sites (Lewis) for adsorbents, whereas coordinatively unsaturated metals or oxygen vacancies act as Lewis acid centers. Bridged hydroxyl groups connecting to transition metals of different valent or oxidation states (e.g., Ce^3+^–O(H)–Ce^4+^) show Brønsted acid character by proton donating ability. Acidity of the samples was characterized by in situ FT-IR spectroscopy using pyridine (Py) adsorption as a base probe molecule. Appendix A shows the Py adsorption spectra of the studied samples. Py adsorption studies supported the strong Lewis acidity of the samples. It was found that only the CeO_2_ sample exhibited some weak Brønsted acidity. The latter result supports the XPS data, detecting reduced Ce^3+^ species on the surface. Spectra of mixed-oxide samples exhibit rather the characteristics of CeO_2_. Considering the amount of acid sites, it can be observed that ZrO_2_ shows a higher acidity compared to CeO_2_. Surface basicity of oxides can be characterized by CO_2_ chemisorption. When basic surface hydroxyl groups react with CO_2_, bicarbonate (hydrogencarbonate) species are formed. Through the interaction of structural O^2−^ ions (Lewis base) with CO_2_, carbonate species can be detected. The various surface species, formed by CO_2_ adsorption, can be observed in the 1800–800 cm^−1^ spectral range [50]. FT-IR spectra of adsorbed CO_2_ on mixed CeO_2_/ZrO_2_ samples are shown in Figure 4. Spectra were collected by room-temperature adsorption for 30 min, followed by desorption at RT (Figure 4A), and at 100 °C (Figure 4B) in high vacuum. Comparing the RT CO_2_ desorption spectra of mixed oxides, similar bands with some variation in intensity ratios can be observed, however the Ce_0.67_Zr_0.33_ sample shows more intensive ones. Characteristic stretching vibrations ν(CO_3_) bands of hydrocarbonates at 1600 and 1410 cm^−1^ [51,52] are very intensive in the sample, and almost diminishes in the other two. This fact is in accordance with XPS investigations, showing a high concentration of OH groups on the CeO_2_ sample and a decreasing tendency with increasing Zr content.

By increasing the desorption temperature to 100 °C, the weakly bound hydrogencarbonate bands disappear and several, better resolved ones appear (Figure 4B). According to Datturi et al. [52], the bands can be associated with surface monodentate (1518 cm^−1^), bidentate (1587, 1346 cm^−1^), and polydentate (1439, 1402 cm^−1^) carbonate species. The other two mixed-oxides with a lower Ce ratio show only monodentate and bidentate carbonate species with much lower intensity. Formation of thermally stable, polydentate, or core-carbonate species indicates the incorporation of carbonate ions into the surface layer, thus its restructuration. It seems that the high mobility of oxygen atoms in pure ceria is stabilized by the presence of zirconium ions and an optimal composition makes the surface more appropriate for CO_2_ adsorption. Daturi et al. [52] draw the conclusion from their similar FT-IR spectroscopic results that the incorporation of Zr^4+^ ions into the ceria framework enhances the surface oxygen relaxation over 50% Ce content and has the opposite effect on the relaxion of Ce^4+^ with already low Zr level, thus stabilizing the surface Ce^4+^ framework. Based on the FT-IR spectroscopic results, it can be expected that Ce_0.67_Zr_0.33_ sample will show more favorable CO_2_ adsorption properties compared to other mixed and pure oxides.

### 3.5. CO_2_ Adsorption 

CO_2_ adsorption breakthrough curves under dynamic conditions of the pure and the mixed-oxide materials are presented in Figure 5. 

CeO_2_ and ZrO_2_ show low adsorption capacity, whereas a significantly higher capacity is detected for the mixed-oxide materials. The shape of the breakthrough curves is similar, which can be due to their similar porosity. The ratio between Ce/Zr influences the adsorption performance of the materials. As predicted by the FT-IR results, the highest capacity for CO_2_ adsorption is detected for Ce_0.67_Zr_0.33_ (3.5 mmol/g) (Table 4). A large quantity of surface oxygen groups or adsorbed O-containing species in Ce–Zr composite nanomaterials could be a reason for the higher CO_2_ capacity. The Ce_0.5_Zr_0.5_ and ext. Ce_0.5_Zr_0.5_ show similar CO_2_ adsorption despite the difference in the surface area.

The observed effect indicates that the surface composition is more important than other structure peculiarities. Moreover, the time required to reach the total adsorption for CeO_2_ and ZrO_2_ materials (T = 17 min) is shorter than that of the mixed-metal oxides (T = 20–26 min). Additional adsorption experiments were performed with the addition of 1 vol.% water vapor to the CO_2_/N_2_ flow at a rate of 30 mL/min to determine the CO_2_ selectivity. In the latter case, higher CO_2_ adsorption capacity was determined for all the adsorbents (Table 4). The highest capacity in the humid environment was detected for Ce_0.67_Zr_0.33_ (3.7 mmol/g). Under dry conditions, surface oxygen reacts with CO_2_ generating carbonate species [14,53]. Under wet conditions, it is assumed that the surface oxygen reacts with water molecules in the gas leading to the formation of hydroxyl groups. The reaction of the hydroxyl groups with CO_2_ results in the generation of hydrogen carbonate species and thereby accelerating the carbon dioxide adsorption.

Adsorption capacity of synthesized nanomaterials were determined using a laboratory scale fixed-bed reactor. The Yoon–Nelson model [54] was applied for adsorption kinetics in a fixed-bed column. The linear form of the model is represented by Equation (1):ln(C_t_/C_o_ − C_t_) = κ_YN_t − τκ_YN_(1)
where k_YN_ is the Yoon–Nelson rate constant (min^−1^), τ is the time required for 50% of adsorbate breakthrough (min), t is the sampling time (min), C_0_ is the initial concentration of CO_2_, and C is the concentration of CO_2_ at any time during evaluation. It was found that the model and experimental data have a strong correlation (R^2^ > 0.99) (Appendix A).

Samples were also tested for CO_2_ capture under equilibrium conditions and the results are presented in Figure 6, Figure 7 and Figure 8. CO_2_ adsorption under static saturation mode without nitrogen stream shows a lower adsorption capacity than those obtained in the dynamic conditions. The heat of adsorption was calculated from CO_2_ adsorption isotherms at 0 °C and 25 °C using the Clausius–Clapeyron equation. They ranged between 25 and 130 kJ/mol. The mixed cerium–zirconium materials exhibited higher values in comparison to CeO_2_ and ZrO_2_ materials_._ The Ce_0.33_Zr_0.67_ and Ce_0.67_Zr_0.33_ (Figure 7) adsorbents demonstrated the highest ones (115−128 kJ/mol) due to the formation of Ce–O–Zr species. They are around three times higher than that of CeO_2_ and ZrO_2_ (40–55 kJ/mol) (Figure 6). Ce_0.5_Zr_0.5_ and ext.Ce_0.5_Zr_0.5_ (Figure 8) have very similar values, which are two times higher than pure oxides (Figure 6). 

Based on the calculated adsorption heats, CO_2_ is physisorbed on CeO_2_ and ZrO_2_ materials whereas chemisorption of CO_2_ is assumed on the mixed cerium–zirconium nanomaterials. According to the literature [14], under dry conditions, the surface oxygen of CeO_2_ reacts with CO_2_, generating carbonate species when the Ce–Zr composites are used as adsorbents. When the mixed Ce/Zr oxides are used as adsorbents, we assume that the stronger interaction of CO_2_ molecules with oxygen of the adsorbent surface is due to the stronger basicity of O^2−^ in them than that of O^2−^ in the pure CeO_2_ and ZrO_2_ nanoparticles.

A stronger interaction between CO_2_ molecules and the Ce–Zr composite materials is also proved by the higher CO_2_ desorption temperature (100 °C), which is needed for total regeneration of the adsorbents.

## 4. Conclusions

CeO_2_, ZrO_2_, and the Ce–Zr composite nanoparticles with a high specific surface area were successfully synthesized using the hydrothermal synthesis procedure. A high CO_2_ adsorption capacity was determined for all the adsorbents depending on their composition and structural peculiarities. Additionally, CO_2_ chemisorption enhanced the CO_2_ capture on Ce–Zr composites due to the presence of strong O^2−^ base sites and enriched surface oxygen species. Materials reused in five adsorption/desorption cycles revealed a high stability with only a slight decrease in adsorption capacity. Among the studied materials, the Ce_0.67_Zr_0.33_ material showed the highest adsorption capacity (3.7 mmol/g). CO_2_ chemisorption is assumed based on the calculated adsorption heat. Enhanced CO_2_ adsorption capacities were detected in experiments with 3 vol.% CO_2_ plus 1 vol.% water vapor due to the additional chemisorption of CO_2_. Total CO_2_ desorption from the Ce–Zr composites was achieved at 100 °C. Experimental data can be appropriately described by the Yoon–Nelson kinetic model. For the first time, it is described that Ce–Zr composite nanomaterials are promising materials for CO_2_ adsorption in dry and humid media.

## Figures and Tables

**Figure 1 nanomaterials-13-02428-f001:**
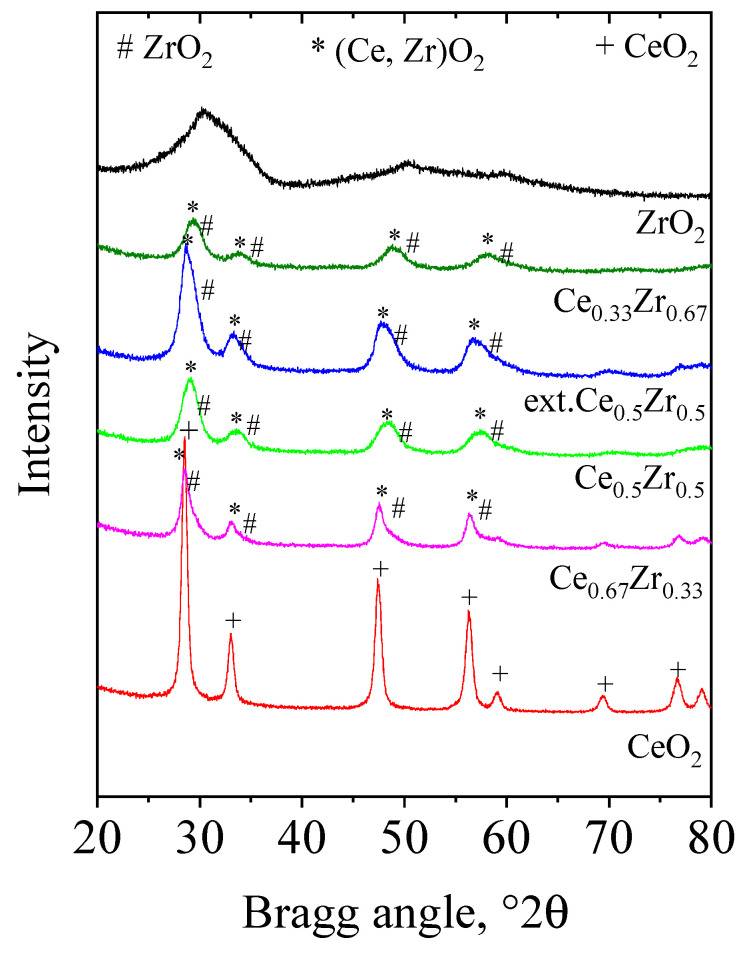
XRD patterns of CeO_2_, ZrO_2_, and Ce-Zr composites (CeO_2_-PDF2 #01-071-4199, ZrO_2_-PDF2 #00-068-0200).

**Figure 2 nanomaterials-13-02428-f002:**
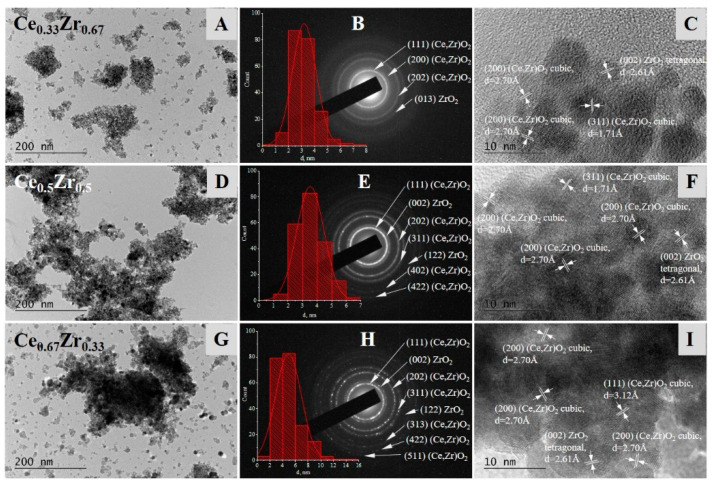
TEM micrographs at low (40,000×) and high (600,000×) magnifications, histograms of the particles size distribution, and Selected Area Electron Diffraction (SAED) patterns of Ce_0.33_Zr_0.67_ (**A**–**C**); Zr_0.5_Ce_0.5_ (**D**–**F**); and Ce_0.67_Zr_0.33_ (**G**–**I**).

**Figure 3 nanomaterials-13-02428-f003:**
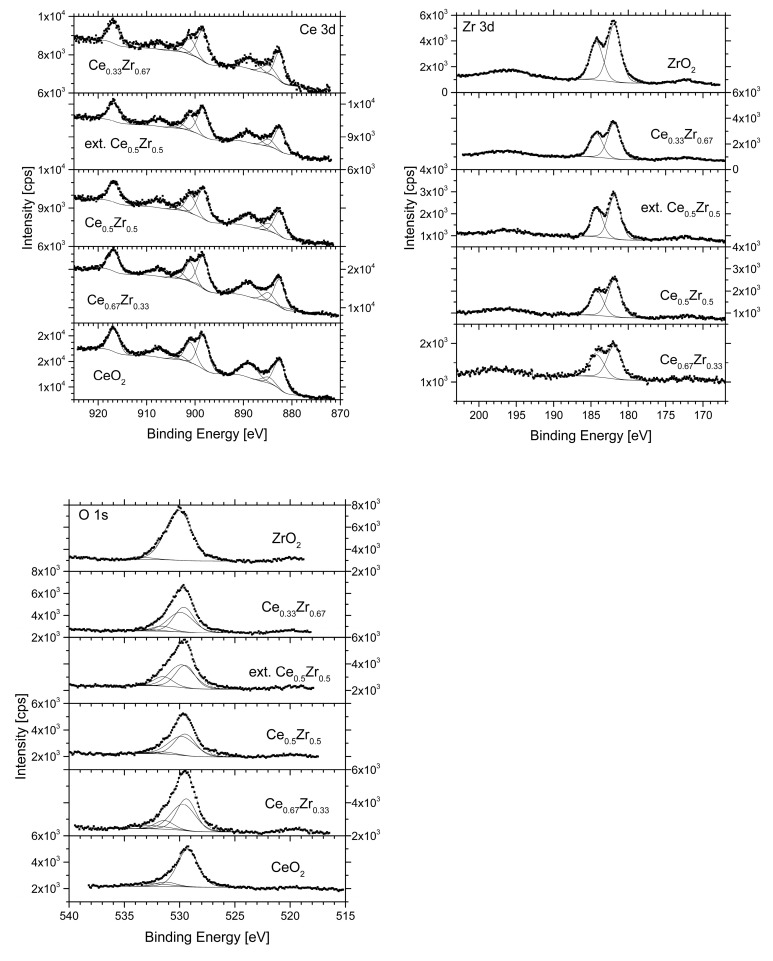
XPS spectra of CeO_2_, ZrO_2_, and CeO_2_/ZrO_2_ samples.

**Figure 4 nanomaterials-13-02428-f004:**
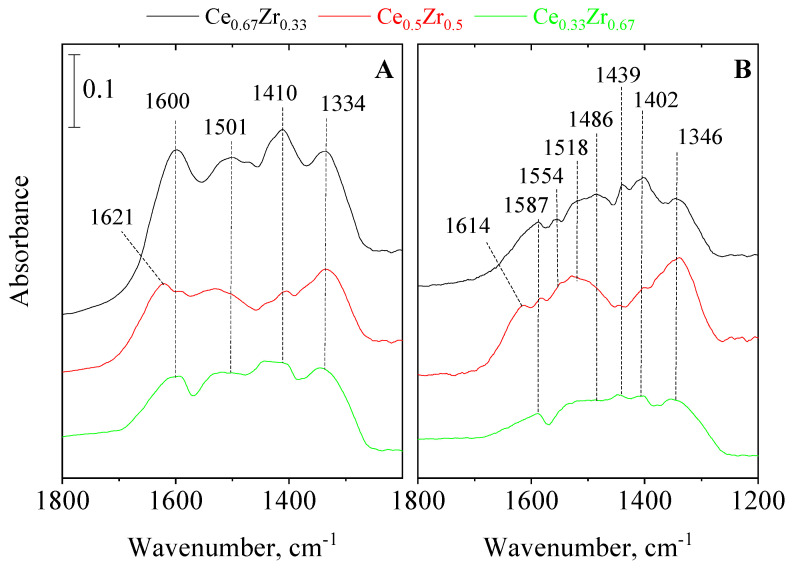
FT-IR spectra of adsorbed CO_2_ on Ce-Zr composites. CO_2_ was adsorbed at RT on 300 °C dehydrated samples followed by RT (**A**) and 100 °C (**B**) desorption in high vacuum.

**Figure 5 nanomaterials-13-02428-f005:**
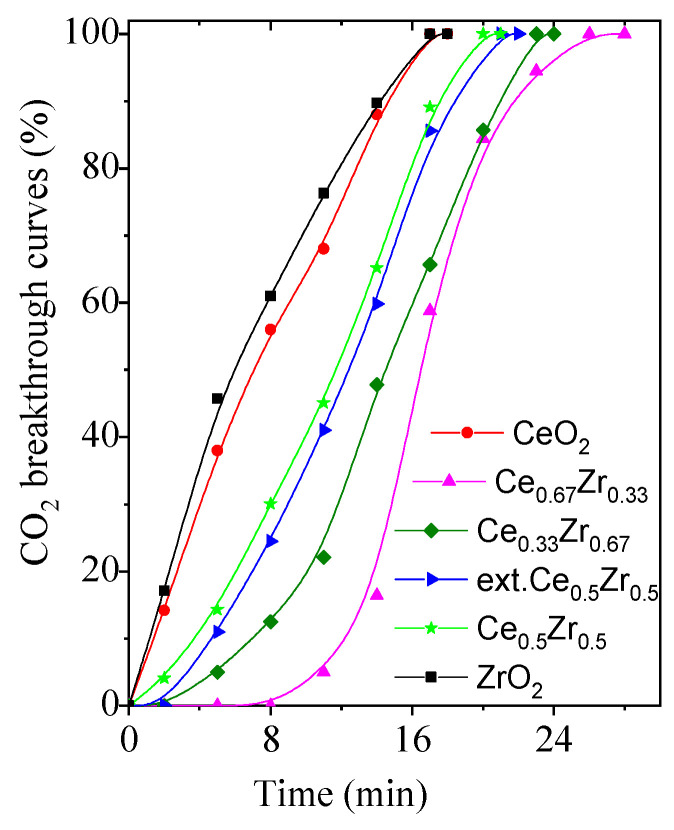
CO_2_ adsorption breakthrough curves of the studied samples.

**Figure 6 nanomaterials-13-02428-f006:**
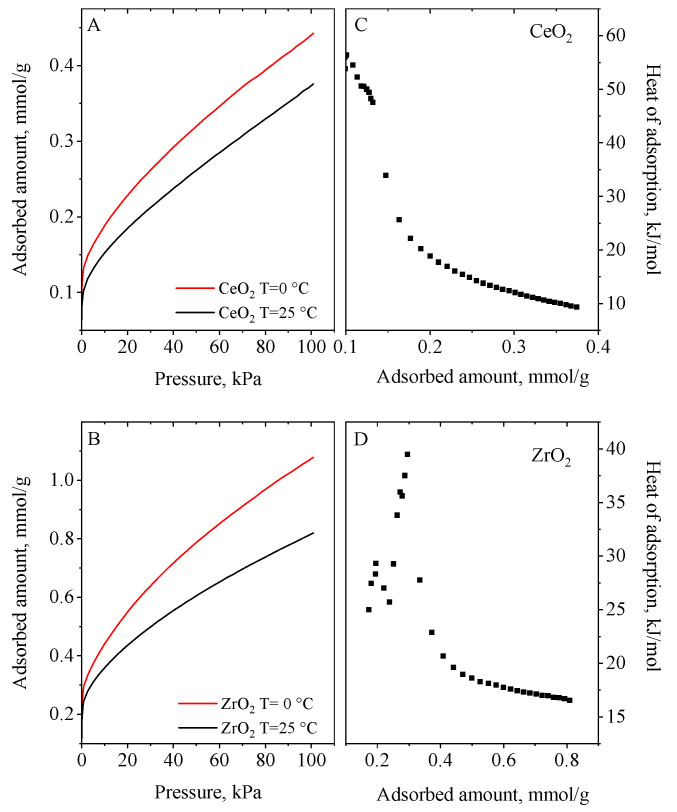
CO_2_ adsorption isotherms at 0 and 25 °C (**A**,**B**) and heat of adsorption curves (**C**,**D**) of the CeO_2_ and ZrO_2_ samples.

**Figure 7 nanomaterials-13-02428-f007:**
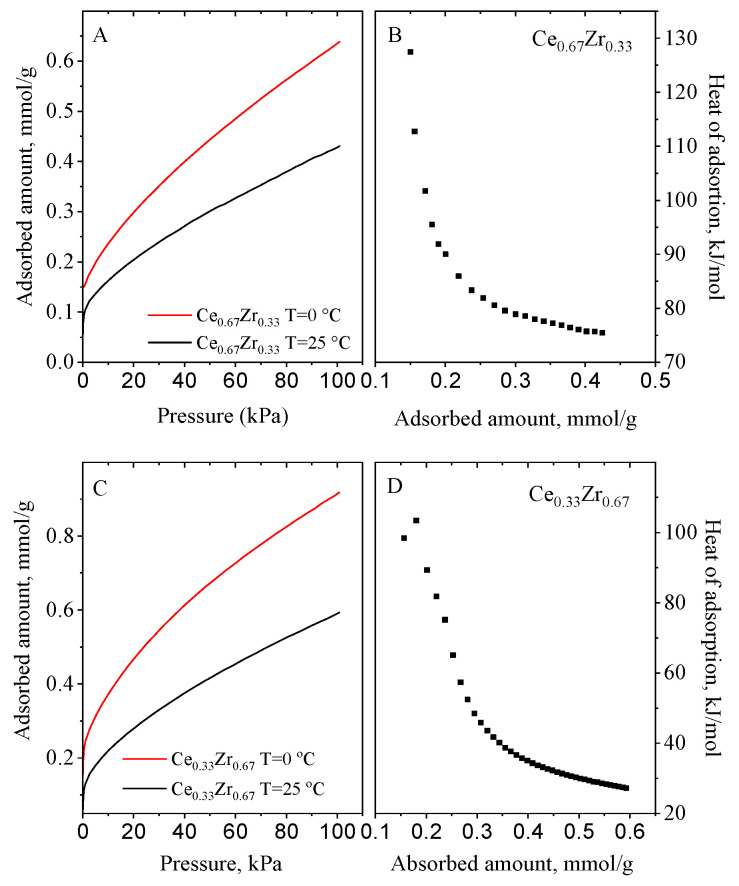
CO_2_ adsorption isotherms at 0 and 25 °C (**A**–**C**) and heat of adsorption curves (**B**–**D**) of the prepared Ce_0.67_Zr_0.33_ and Ce_0.33_Zr_0.67_ composites.

**Figure 8 nanomaterials-13-02428-f008:**
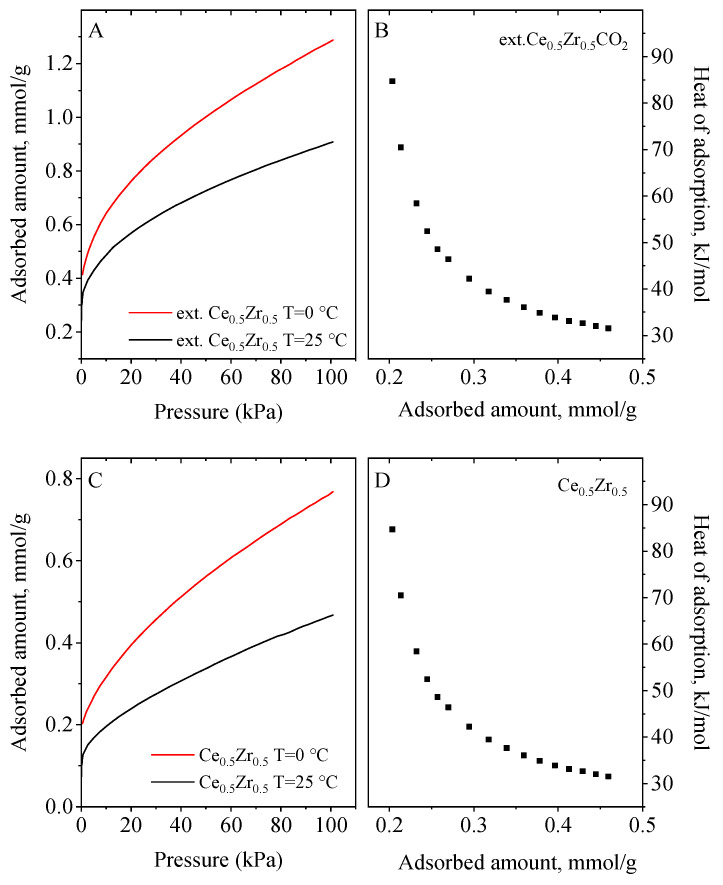
CO_2_ adsorption isotherms at 0 and 25 °C (**A**–**C**) and heat of adsorption curves (**B**–**D**) of the prepared Ce_0.5_Zr_0.5_ and Ce_0.5_Zr_0.5_ composites.

**Table 1 nanomaterials-13-02428-t001:** Textural properties of cerium- and zirconium-oxide materials.

Samples	S_BET_, m^2^/g	Pore Volume, cm^3^/g
ZrO_2_	271	0.41
Ce_0.33_Zr_0.67_	185	0.18
ext.Ce_0.5_Zr_0.5_	181	0.15
Ce_0.5_Zr_0.5_	153	0.17
Ce_0.67_Zr_0.33_	119	0.13
CeO_2_	55	0.10

**Table 2 nanomaterials-13-02428-t002:** Phase composition, fluorite unit cell parameters, and average crystallite size of CeO_2_, ZrO_2_ and the Ce–Zr composite nanomaterials.

Samples	Phase Composition (Space Group)	Unit Cell Parameters (Å)	Crystallite Size, nm (±0.5–1 nm)
CeO_2_	CeO_2_ (Fm-3m)	a = 5.4146(3)	16
Ce_0.67_Zr_0.33_	(Ce,Zr)O_2_ (Fm-3m) ZrO_2_ (P42/nmc)	a = 5.406(1) a = 3.709(5) c = 5.33(1)	12 5
Ce_0.5_Zr_0.5_	(Ce,Zr)O_2_ (Fm-3m) ZrO_2_ (P42/nmc)	a = 5.382(5) a = 3.710(5) c = 5.313(8)	8 6
ext.Ce_0.5_Zr_0.5_	(Ce,Zr)O_2_ (Fm-3m) ZrO_2_ (P42/nmc)	a = 5.394(2) a = 3.709(3) c = 5.339(5)	10 5
Ce_0.33_Zr_0.67_	(Ce,Zr)O_2_ (Fm-3m) ZrO_2_ (P42/nmc)	a = 5.36(1) a = 3.70(1) c = 5.31(2)	6 5
ZrO_2_	n.d.	n.d.	n.d.

**Table 3 nanomaterials-13-02428-t003:** Surface composition of the CeO_2_, ZrO_2_, Ce_0.67_Zr_0.33_, Ce_0.33_Zr_0.67_, and Ce_0.5_Zr_0.5_ materials based on XPS analysis.

Samples	Concentration, at. %			Ratio of Oxidation States, %	Binding Energy, eV	
	Ce (Ce 3d)	Zr (Zr 3d)	O (O 1s)	Ce^4+^:Ce^3+^	O1s Ce^4+^	O1s Zr
CeO_2_	27	-	73	93:7	529.27	-
Ce_0.67_Zr_0.33_	17	12	71	87:13	529.39	529.68
Ce_0.5_Zr_0.5_	11	23	67	91:9	529.49	529.79
ext.Ce_0.5_Zr_0.5_	10	21	69	90:10	529.49	529.79
Ce_0.33_Zr_0.67_	5	28	67	87:13	529.61	529.90
ZrO_2_	-	37	63	-	-	530.09

**Table 4 nanomaterials-13-02428-t004:** CO_2_ adsorption capacities of the prepared materials in dynamic conditions.

Samples	CO_2_ ads. CO_2_/N_2_, mmol/g	CO_2_ ads. CO_2_/H_2_O/N_2_ ^1^, mmol/g	Repeated CO_2_ ads. CO_2_/H_2_O/N_2_ ^2^, mmol/g
ZrO_2_	2.0	2.1	1.9
Ce_0.33_Zr_0.67_	3.2	3.4	3.2
Ce_0.5_Zr_0.5_	2.9	3.1	3.0
ext.Ce_0.5_Zr_0.5_	3.0	3.5	3.4
Ce_0.67_Zr_0.33_	3.5	3.7	3.6
CeO_2_	2.3	2.4	2.3

^1^ Experiments performed with 3 vol.% CO_2_ plus 1 vol.% water vapor; ^2^ results obtained by 5 adsorption/desorption cycles.

## Data Availability

Not applicable.

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
