# Peer review of "Impact of Ce/Zr Ratio in the Nanostructured Ceria and Zirconia Composites on the Selective CO2 Adsorption"

_nanomaterials, 2023, doi:10.3390/nano13172428_

Round 1
Reviewer 1 Report
In this article, the authors investigated the effect of different Ce/Zr ratios on CO2 adsorption. The study is of some significance. They demonstrated that the strong CO2 adsorption capacity is mainly attributed to the formation of O2- base sites and abundant surface oxygen species on the material surface. This article is suitable for publication in the journal.
The following are my concerned questions:
1: I noted that the measured ratio of Ce/Zr on the surface is far lower than the theoretical one. Can the authors explain that?
2. Is the acidic properties obtained by IR measurements related with adsorption of CO2? The author would better discuss this.
Author Response
The authors thank the reviewer for the comprehensive review. The suggestions have been accepted and were taken into consideration. The needed corrections have been made and a corrected version of the manuscript has been submitted.
Remark 1: I noted that the measured ratio of Ce/Zr on the surface is far lower than the theoretical one. Can the authors explain that?
Answer: XPS analysis data of high ceria containing samples (CeO2, Ce0.67Zr0.33) show that the concentration of Ce is lower on the surface than theoretically calculated, whereas the oxygen content is higher than the theoretical content (73/71 at. %, instead of 66 at. %). It means that these samples have surface rich with defect sites and OH groups due to the very small particle size or adsorbed O-containing species. The Ce3+ surface concentration is higher for Ce-Zr composite materials as compared to pure CeO2 (Table 3) which indicates high degree of Zr incorporation into ceria lattice with formation of oxygen vacancies. The manuscript was corrected accordingly.
Remark 2: Is the acidic properties obtained by IR measurements related with adsorption of CO2? The author would better discuss this.
Answer1: The surface species of oxides can be described as acid-base pairs. Oxygen atoms of the structure serve as Lewis basic sites, whereas coordinatively unsaturated metals or oxygen vacancies act as Lewis acid centers. Mixed-ceria-zirconia oxides show acidic and basic surface properties as evidenced formerly by adsorption of different probe molecules by Daturi et.al (Phys.chem. Chem Phys. 1999, 1 5717-5724), and J. I. Gutiérrez-Ortiz (Journal of Thermal Analysis and Calorimetry, Vol. 80 (2005) 225–228). The acidic properties of the adsorbents are also part of the overall picture, however, we acknowledge that CO2 adsorption is more relevant in our case. The Py adsorption FT-IR figure was moved to Supporting Data, and the manuscript was shortened in this part.
Reviewer 2 Report
1. In the introduction, it is necessary to indicate the literature temperature values for CO2 adsorption and desorption for CeO2, ZrO2 and mixed oxide Ce/Zr nanoparticles.
2. It is necessary to point to the literature data on the adsorption and desorption of CO2 for other oxides, in particular: CaO and others, for example, https://doi.org/10.1016/j.ceramint.2021.11.296, https://doi. org/10.1016/j.jcou.2022.102353, https://doi.org/10.1016/j.jcou.2023.102517, https://doi.org/10.1134/S1070363222020232 and others.
3. Ce/Zr chlorides were used according to the synthesis procedure. However, the experimental procedure does not indicate whether the Ce/Zr hydroxides were washed to remove residual chlorine. It should be noted that Cl- strongly influences the mechanism of processes in the autoclave.
4. In fig. 2. You must specify the PDF card number for CeO2, ZrO2.
5. How do the average crystallite sizes for CeO2, ZrO2 and mixed oxide Ce/Zr nanoparticles compare according to XRD and electron microscopy data.
The work "Impact of Ce/Zr ratio in the nanostructured ceria and zirconia mixed oxides on the selective CO2 adsorption" is written in good scientific language and may be of interest to a wide range of researchers.
Minor editing of English language required
Author Response
The authors thank the reviewer for the comprehensive review. The suggestions have been accepted and were taken into consideration. The needed corrections have been made and a corrected version of the manuscript has been submitted.
Remark 1: In the introduction, it is necessary to indicate the literature temperature values for CO2 adsorption and desorption for CeO2, ZrO2 and mixed oxide Ce/Zr nanoparticles.
Answer: The needed information for CeO2 is added in the MS. However, there is no information about CO2 adsorption for ZrO2 and mixed oxide Ce/Zr nanoparticles.
Remark 2: It is necessary to point to the literature data on the adsorption and desorption of CO2 for other oxides, in particular: CaO and others, for example, https://doi.org/10.1016/j.ceramint.2021.11.296, https://doi. org/10.1016/j.jcou.2022.102353, https://doi.org/10.1016/j.jcou.2023.102517, https://doi.org/10.1134/S1070363222020232 and others.
Answer: The papers are added in the Reference list and are discussed in the MS.
Remark 3: Ce/Zr chlorides were used according to the synthesis procedure. However, the experimental procedure does not indicate whether the Ce/Zr hydroxides were washed to remove residual chlorine. It should be noted that Cl- strongly influences the mechanism of processes in the autoclave.
Answer: The samples were washed with distilled water. We apologize for the mistake and the needed corrections are made.
Remark 4: In fig. 2. You must specify the PDF card number for CeO2, ZrO2.
Answer: The pdf cards are added in the MS.
Remark 5: How do the average crystallite sizes for CeO2, ZrO2 and mixed oxide Ce/Zr nanoparticles compare according to XRD and electron microscopy data.
Answer: On the basis of XRD patterns of the samples, the mean coherent domain size is calculated for all of the crystalline phases present in the sample – namely the fluorite type (Ce,Zr)O2 and the tetragonal ZrO2. For samples with the presence of zirconia, it is supposed that the ZrO2 phase contains a fraction of crystalline but very small particles, i.e. around 5-6 nm. The peaks from such fraction will be very broad and they could not be distinguished from the background of the diffraction pattern. The effect is most pronounced for the ZrO2 sample and is typical for XRD amorphous material and phase parameters and crystallite size cannot be determined. By means of TEM images, all registered nanoparticles (even those with size below 3 nm) are measured and their mean size is calculated, regardless of the phase. The different approach to the mean size determination in the case of the two methods – XRD and TEM is the reason for the slightly different values of the nanoparticle sizes obtained.
Reviewer 3 Report
The manuscript titled “Impact of Ce/Zr ratio in the nanostructured ceria and zirconia mixed oxides on the selective CO2 adsorption” presented by Gloria Issa, Martin Kormunda, Oyundari Tumurbaatar, Ágnes Szegedi, Daniela Kovacheva, Daniela Karachanova, and Margarita Popova. However, there are major issues that could be clarified:
- In the abstract: “High surface area, mesoporous CeO2, ZrO2 and mixed oxide Ce/Zr nanoparticles” sounds confusedly. How nanoparticles could be high surface area and mesoporous?
- In section 2.3 the description of XPS has some incorrect terms. The sentence “An achromatic X-ray source XR50 (SPECS) was used with an achromatic Al X-ray tube and Kα line (energy of 1 486.6 eV) at 12 kV, 200 W, no flood gun was used.” should be corrected like that: “The spectrometer is equipped with a non-monochromatic X-ray source XR50 with double Al/Mg anode operated under 12 kV (200W), the core-level spectra were measured using Al Kα radiation (hν = 1486.6 eV ), no flood gun was used.”
- “In-situ” should be written correctly “in situ”.
- BET-measurements should be presented only in the table format. Please, move figure 1 to the supporting information.
- Please, mark the peaks (and corresponded phases) observed in the XRD patterns of the samples under study (Figure 2). What labels 1Ce0.5Zr0.5 and 5Ce0.5Zr0.5 mean (Fig. 2)?
- Which peak was used for estimation of crystallite size for both phases?
- There are some contradictions between the XRD and TEM analysis. It looks that the particles size distribution has been measured for TEM images with low resolution (200 nm scale at Fig.3A, D, G) and it leads to decrease in mean particle size for the samples. But on Fig.3C, F, I one can see the particles with size close to the crystallite size measured by XRD. Please, add to SI the images with high resolution and calculate the particles size distribution for them. The statement “ZrO2 was not registered by TEM, probably due to its more amorphous structure” sounds confusedly, in case when it was detected by XRD (as authors considered). Was ZrO2 detected by XRD for Ce/Zr samples or not? As I mentioned above to clarify this it is necessary to mark all peaks observed by XRD.
- In the XPS section authors discuss the O1s core-level spectra that are not presented in the main text but did not discuss the presented Zr3d core level spectra. Please, provide O1s spectra and add the description of Zr3d core-level spectra.
- The statement “The absence of zirconium oxide in a lower oxidation state is indicative for segregation of the zirconia phase over the ceria particles, which is in accordance with XRD and TEM results.” First, of zirconium oxide could not be in any oxidation state, zirconium cations could be. Second, Absence of any phase (or lower oxidation state of cation) does not indicate the segregation this phase over the other phase. XRD and TEM do not present any segregation processes. An XPS is not structure-sensitive method.
- The authors use unsuitable term for their system, for example, mixed Ce/Zr nanoparticles (Conclusion). In fact, the adsorbents consist of ZrO2 (XRD), CeO2 (TEM), CeZrO4 phase (TEM), and the other amorphous phases could be detected. I recommend use term, for example, Ce-Zr composite adsorbent.
The current version of manuscript demands some revision and corrections.
English should be revised.
Author Response
Please, see the attached file.

Round 2
Reviewer 3 Report
The manuscript hold be accepted for publication
Ok